# TacLoc: Global Tactile Localization on Objects from a Registration Perspective

Zirui Zhang, Fumin Zhang and Huan Yin

*Abstract*—**Existing tactile-based methods generally rely on rendering data or pre-trained models, which limits generalizability and efficiency. In this study, we propose TacLoc, a novel tactile localization framework that addresses these challenges by formulating the problem as a one-shot point cloud registration task. TacLoc introduces a graph-theoretic partial-to-full registration method, leveraging tactile sensor-generated dense point clouds and surface normals for efficient and accurate pose estimation. The approach eliminates the need for extensive training data. It achieves robust performance through normal-guided graph pruning and a hypothesis-and-verification pipeline. Evaluations on the YCB-based dataset demonstrate the superiority of TacLoc in terms of accuracy, efficiency, and generalization. We also perform the real-world tests using Gelsight Mini sensor.**

## I. INTRODUCTION

Tactile sensing presents a promising strategy for contact-based sensing in robotic manipulation, particularly for tasks subject to visual occlusion. A crucial initial step for successful manipulation is accurately determining the robotic manipulator's initial pose relative to an object. Current tactile localization methods often employ Monte Carlo Localization (MCL) [1], [2] or brute-force search within the $\mathfrak{se}(3)$ space [3], [4]. These techniques estimate pose by matching real sensor data against a pre-rendered or simulated model, a process that can be computationally demanding and may necessitate extensive training. In this study, we introduce TacLoc, a novel tactile localization framework that surpasses point cloud registration methods in accuracy. Utilizing a hypothesis-and-verification scheme, TacLoc obviates the need for sequential Bayesian inference. Instead, TacLoc identifies the optimal transformation by minimizing point-to-plane residuals in a single, one-shot localization. It achieves a mean rotation error (RE) of 22.38°, translation error (TE) of 5.0mm (for a total movement of 10mm) in TACTO simulator [5] when tested on 10 YCB objects [6]. We also demonstrate its application on a real 3D-printed object (Figure 1).

## II. METHOD

**From Raw Data to Initial Correspondence.** As depicted in Figure 2, the process begins with the conversion of raw tactile images into points and normals. A point cloud submap can then be constructed from a sequence of these measurements, leveraging end-effector poses. Alternatively, a single-shot measurement can be used for one-shot pose estimation.

To ensure computational efficiency during pose estimation, voxel grid downsampling is applied to the point clouds. Key-

All authors are with the Cheng Kar-Shun Robotics Institute, Hong Kong University of Science and Technology, Hong Kong SAR.

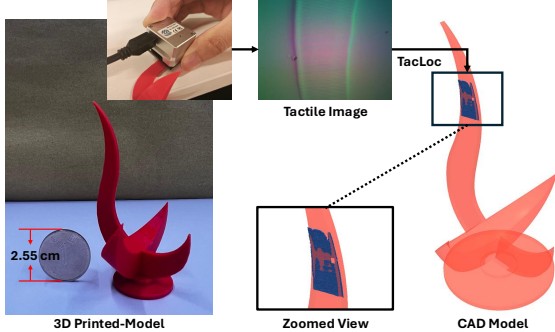

Fig. 1. Real-world demonstration. TacLoc performs fast, accurate, and robust 6DoF object localization based on only touch sensing.

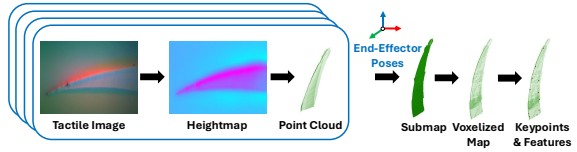

Fig. 2. For each tactile frame, we convert it to a point cloud with normals and build a submap using end-effector poses. This submap is then processed by our front-end pipeline.

points are then detected using the Intrinsic Shape Signatures (ISS) and are encoded FPFH for each keypoint. Finally, initial correspondences are established by performing Manhattan distance matching in the feature space.

**Multiple Pose Hypotheses Generation.** A compatibility graph is constructed based on pairwise geometric consistency check, including distance, normal alignment, and injective correspondence checks. In this graph, nodes represent individual correspondences, while edges connect pairs of correspondences deemed mutually compatible. Maximal cliques within this grasph are subsequently extracted and ranked by size using a modified Bron-Kerbosch algorithm [7]. The top $K$ cliques are selected to estimate the pose.

For each selected clique, we estimate its transformation $\mathbf{T}_k = (\mathbf{R}_k, \mathbf{t}_k)$ by minimizing both point-to-point and normal-to-normal residuals. The rotation component is estimated as:

$$\mathbf{R}_k = \arg\min_{\mathbf{R}_k} \sum_{(i,j) \in C_k} \left( \|\mathbf{q}'_j - \mathbf{R}_k \mathbf{p}'_i\|^2 + \alpha \cdot \arccos^2 \langle \mathbf{m}_j, \mathbf{R}_k \mathbf{n}_i \rangle \right)$$

(1)

where $\mathbf{p}'_i$ and $\mathbf{q}'_j$ represent centered points from the source and target clouds, respectively; $\mathbf{n}_i$ and $\mathbf{m}_j$ denote their corresponding normals. The term $\alpha$ balances the weight between distance and normal differences. The translation component is

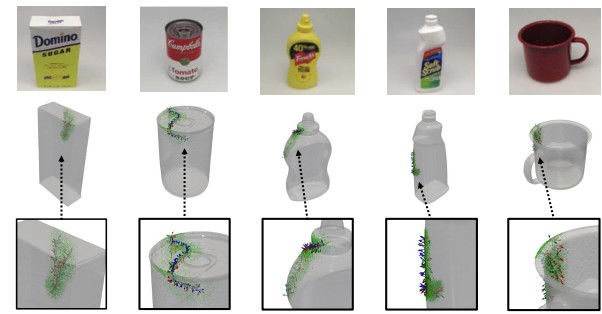

Fig. 3. TacLoc aligns a tactile-sensed point cloud to a CAD model by finding the transformation that best fits the points to the model's surfaces, minimizing point-to-plane error.

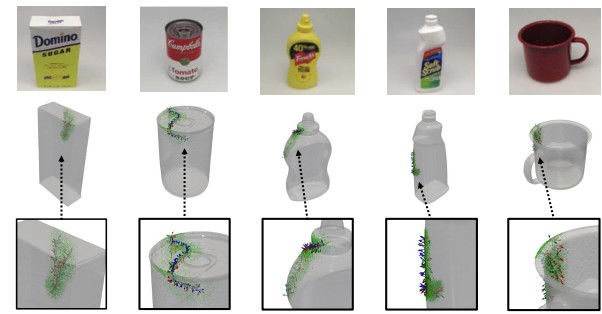

Fig. 4. Selected samples from the YCB-Reg benchmark. Gray semi-transparent objects represent the target models, while green point clouds correspond to tactile-based point clouds using the pre-processing approach. Zoomed views of these sliding samples are also presented at the bottom.

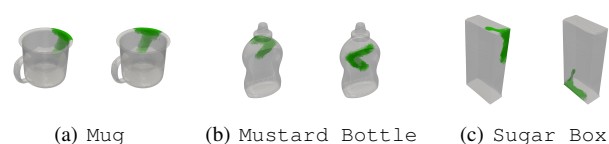

(a) Mug    (b) Mustard Bottle    (c) Sugar Box

Fig. 5. Three failure case studies. In each subfigure, the left shows the ground truth while the right displays the misalignment.

TABLE I
6 DoF LOCALIZATION RESULTS

| Method | RE (°) | TE (mm) | RR (%) | Time (s) |
|---|---|---|---|---|
| RANSAC [8] | 127.89 | 99.57 | 1.0 | **1.74** |
| TEASER++ [9] | 65.92 | 12.24 | 46.0 | 12.24 |
| 3DMAC [10] | 58.43 | 31.65 | 46.0 | 23.92 |
| TacLoc (Ours) | **22.38** | **5.00** | **85.0** | 3.16 |

then estimated following:

$$\mathbf{t}_k = \arg\min_{\mathbf{t}_k} \sum_{(i,j)\in C_k} \|\mathbf{t}_k - (\mathbf{q}_j - \mathbf{R}_k\mathbf{p}_i)\|^2 \qquad (2)$$

Finally, we generate $K$ pose candidates based on the pruned correspondences, which are obtained from the maximal cliques used for multiple hypotheses generation.

**Pose Verification and Refinement.** We utilize a point-to-plane loss function for geometric verification and refinement. This function, which is based on spatial proximity rather than feature descriptors, is expressed as follows:

$$\mathcal{L}(\delta\mathbf{R}_k, \delta\mathbf{t}_k) = \sum_i \langle \delta\mathbf{R}_k\hat{\mathbf{p}}_i^{\text{src}} - \tilde{\mathbf{q}}_i + \delta\mathbf{t}_k, \tilde{\mathbf{m}}_i \rangle^2 \qquad (3)$$

where $\hat{\mathbf{p}}_i^{\text{src}} = \mathbf{R}_k\mathbf{p}_i^{\text{src}} + \mathbf{t}_k$, $\tilde{\mathbf{q}}_i$ denotes the closest point to the transformed source point $\hat{\mathbf{p}}_i^{\text{src}}$ in the downsampled target cloud, and $\tilde{\mathbf{m}}_i$ represents the associated normal of $\tilde{\mathbf{q}}_i$. The refined solution $\mathbf{T}_k^* = (\delta\mathbf{R}_k^*\mathbf{R}_k, \delta\mathbf{R}_k^*\mathbf{t}_k + \delta\mathbf{t}_k^*)$ is regarded as the hypothesis $\boldsymbol{\theta}_k$ for clique $C_k$. And the weight is granted by $w_k = \exp[-\mathcal{L}(\delta\mathbf{R}_k^*, \delta\mathbf{t}_k^*)]$. The transformations achieving lower residual errors receive higher weights, with the highest-scoring estimate selected as the final pose $\mathbf{T}^*$.

## III. EXPERIMENTS AND RESULTS

**YCB-Reg Benchmark Comparison.** We evaluate our global localization method, TacLoc, on the YCB-Reg tactile dataset (Figure 4) against three baselines: RANSAC [8], TEASER++ [9], and 3DMAC [10]. The dataset consists of ten YCB objects [6], with ten $10cm$ slides per object collected via the TACTO simulator [5].

TacLoc's hypothesis-and-verification framework effectively handles ambiguous cases through a weighted consensus of multiple transformation candidates. Results are reported in Table I, including average rotation error (RE), translation error (TE), recall rate (RR) under 10mm and 30°, and execution time. Figure 5 illustrates three representative failure cases, attributed to repetitive features that introduced ambiguity during registration.

**Real-world Demonstration.** The proposed TacLoc is further demonstrated using a GelSight Mini sensor on a 3D-printed model. The target point cloud is generated by uniformly sampling the original CAD model, shown in Figure 1. For the measurements, the GelSight SDK provides the heightmap, gradient map, and contact mask. Subsequently, the point cloud and surface normals are derived from the reconstructed geometry.

To ensure thorough validation, the demonstration was repeated ten times on different regions of the model. Three key challenges were identified: a *single, non-sliding touch* was used, which is more difficult than sliding-based methods; the 3D-printed model was highly symmetrical and lacked texture; and minor manufacturing defects caused discrepancies between the CAD and physical models. Under these challenging conditions, we report tht the success rate for the real-world TacLoc was 6/10.

Figure 6 shows the registration results. Figure 6(a) depicts a successful case where the sensor captured the body's curvature without reaching the edge, while Figure 6(b) shows a failure where the sensor only detected the outer curve, missing the inner curved edge.

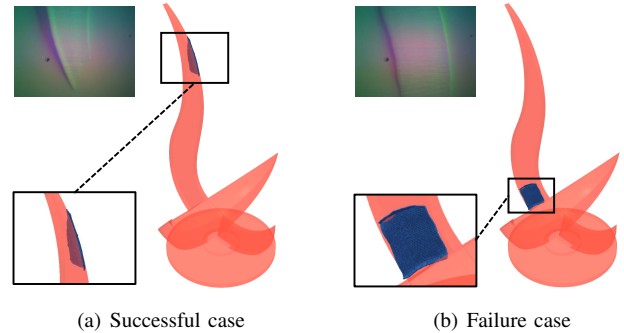

(a) Successful case    (b) Failure case

Fig. 6. Real-world demonstration is conducted on a 3D-printed object, where only a single touch is performed without sliding. The proposed TacLoc method is employed for one-shot global tactile localization to estimate the pose.

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
