# OpenReview forum: "TacLoc: Global Tactile Localization on Objects from a Registration Perspective"
_IEEE.org/IROS/2025/Workshop/Tactile_Sensing — IROS 2025 Workshop Tactile Sensing Poster_

### Official Review · Reviewer_usnW · 2025-09-23
**Review for submission #10**

**Rating:** 7
**Confidence:** 4

**Review:**

The paper proposes a localization method using a vision-based tactile sensor and end-effector poses. Specifically, the authors first extract a height map and a point cloud from the tactile data, then convert them to a 3D voxelized map by leveraging the end-effector pose, and finally extract keypoints and spatial features to localize it within a given CAD model.

Strengths:
* The method is clearly presented with illustrative figures.
* The proposed method is pretty efficient (reasonably low runtime with low error).
* The method is tested in both simulation and a real-world setup.
* The paper is well written and easy to understand.

Questions/Weakness:
* The real-world testing is executed on a 3D printed object, which as claimed in the paper is highly-symmetrical and textureless. It would be more convincing to test on multiple real objects, inside and outside YCB dataset.
* The baseline comparison could be more convincing if comparing with more recent methods like "MidasTouch: Monte-Carlo inference over distributions across sliding touch" or visual-tactile methods like "Active Visuo-Tactile Point Cloud Registration for Accurate Pose Estimation of Objects in an Unknown Workspace". A short justification of why using tactile sensing only can be helpful too.

---

### Official Review · Reviewer_FdA9 · 2025-09-23

**Rating:** 7
**Confidence:** 4

**Review:**

TacLoc presents a high-quality and clearly articulated contribution to tactile-based object localization by reformulating it as a one-shot point cloud registration problem. Its originality lies in a graph-theoretic, hypothesis-and-verification pipeline that requires no pre-training, offering significant advantages in generalizability and computational efficiency over methods reliant on rendered data or sequential inference. The work is substantiated by rigorous benchmarking on the YCB-Reg dataset and promising real-world tests with a GelSight sensor. Key strengths include its strong accuracy, efficiency, and model-agnostic design. However, the approach shows limitations in handling objects with repetitive features or high symmetry, and its real-world performance can be impacted by minor discrepancies between CAD models and physical objects, as evidenced by a a bit low success rate under challenging single-touch conditions.